# Rapid Determination of Nitrate Nitrogen Isotope in Water Using Fourier Transform Infrared Attenuated Total Reflectance Spectroscopy (FTIR-ATR) Coupled with Deconvolution Algorithm

**DOI:** 10.3390/molecules28020567

**Published:** 2023-01-05

**Authors:** Ke Wu, Fei Ma, Cuilan Wei, Fangqun Gan, Changwen Du

**Affiliations:** 1College of Environment and Ecology, Jiangsu Open University, Nanjing 210017, China; 2The State Key Laboratory of Soil and Sustainable Agriculture, Institute of Soil Science Chinese Academy of Sciences, Nanjing 210008, China; 3College of Advanced Agricultural Sciences, University of Chinese Academy of Sciences, Beijing 100049, China

**Keywords:** nitrate, nitrogen isotope, FTIR-ATR, deconvolution algorithm, *PLSR* model

## Abstract

Nitrate is a prominent pollutant in water bodies around the world. The isotopes in nitrate provide an effective approach to trace the sources and transformations of nitrate in water bodies. However, determination of isotopic composition by conventional analytical techniques is time-consuming, laborious, and expensive, and alternative methods are urgently needed. In this study, the rapid determination of ^15^NO_3_^−^ in water bodies using Fourier transform infrared attenuated total reflectance spectroscopy (FTIR-ATR) coupled with a deconvolution algorithm and a partial least squares regression (*PLSR*) model was explored. The results indicated that the characteristic peaks of ^14^NO_3_^−^/^15^NO_3_^−^ mixtures with varied ^14^N/^15^N ratios were observed, and the proportion of ^15^NO_3_^−^ was negatively correlated with the wavenumber of absorption peaks. The *PLSR* models for nitrate prediction of ^14^NO_3_^−^/^15^NO_3_^−^ mixtures with different proportions were established based on deconvoluted spectra, which exhibited good performance with the ratio of prediction to deviation (*RPD*) values of more than 2.0 and the correlation coefficients (*R*^2^) of more than 0.84. Overall, the spectra pretreatment by the deconvolution algorithm dramatically improved the prediction models. Therefore, FTIR-ATR combined with deconvolution and *PLSR* provided a rapid, simple, and affordable method for determination of ^15^NO_3_^−^ content in water bodies, which would facilitate and enhance the study of nitrate sources and water environment quality management.

## 1. Introduction

In recent years, with the gradual control of fixed source pollution, agricultural non-point source pollution has become the main factor affecting the water quality. According to the Bulletin of the Second National Survey of Pollution Sources in 2017, the total nitrogen discharge of water pollutants in China was 3.014 million tons and the total phosphorus discharge was 31.54 million tons, of which the total nitrogen discharge of agricultural water pollutants was 1.41 million tons, and the total phosphorus discharge was 21.2 million tons, accounting for 46.52% and 67.22% of the total, respectively [1]. The agricultural sources become the main contributors to total nitrogen and total phosphorus discharges. Compared with phosphorus, the form of nitrogen, including organic nitrogen and inorganic nitrogen, is much more complicated, Ammonium nitrogen (NH_4_^+^-N), nitrate nitrogen (NO_3_^−^-N) and nitrite nitrogen (NO_2_^−^-N) are the main forms of inorganic nitrogen in water. Nitrate nitrogen is an important form of active nitrogen that can cause deterioration of aquatic ecosystem health, i.e., eutrophication, and human health risks; the former included the items of decreased oxygen levels in water bodies, algal blooms, and reduced aquatic biodiversity, and the latter included the items of stomach cancer, diabetes, thyroid disease, and “blue baby” syndrome [2,3,4]. The nitrate contents in water bodies are regulated by the interaction between external sources of pollution (e.g., industrial wastewater, municipal sewage, nitrogen runoff, and atmospheric deposition) and nitrogen cycling processes (e.g., nitrification, denitrification, ammonia volatilization, and plant/microbial utilization) [5,6,7]. As one of the main means to evaluate the degree of impact of agricultural non-point source pollutants on water environment quality, identification and determination of the main sources and transformations of nitrate are the primary objectives to construct effective strategies to address nitrate pollution.

The nitrate isotope (^15^N/^18^O-NO_3_) provides an effective method to determine the potential nitrate sources and transformations [8,9]. The advantages of the isotope method include direct identification, less auxiliary information, and high precision in identifying pollution sources. Therefore, rapid, accurate, and affordable detection techniques of isotopic composition are extremely important. Usually, an isotope ratio mass spectrometer (IRMS) is a necessary instrument for measuring stable isotopic composition [10]. However, the IRMS is limited for many organizations and institutions due to its high price and maintenance costs. In addition, sample pretreatments, including ion exchange, cadmium azide reduction, and bacterial denitrification, are time- and labor-consuming. Given these issues, stable isotopes of nitrogen tend to be limited in practical applications. Therefore, it is necessary to develop an economical and effective method for the isotopic content qualification of water bodies.

As a nondestructive, rapid, and efficient analysis method, Fourier transform infrared attenuated total reflectance spectroscopy (FTIR-ATR) has been widely used in various fields. Due to the characteristic absorption peaks of nitrate around 1360 cm^−1^, the contents of nitrate in soil, vegetables, and water bodies have been quantitatively determined by FTIR-ATR [11,12,13,14,15]. In comparison to the spectral absorption of ^14^NO_3_^−^, the absorption band of ^15^NO_3_^−^ significantly shifted to the direction of low wavenumber, which has already been used to determine nitrogen isotope-labelled nitrates in solution and soil using the *PLSR* model [16,17,18]. Since the absorption of nitrate spectra is prone to strong interference from water, the previous studies eliminated the interference by directly deducting water absorption, and the *PLSR* models based on water yielded a determination error of 6.7–9.2 mg/L [16]. However, such a water deduction algorithm resulted in poor prediction performance in low concentrations with different N-isotopically labeled nitrates due to the relatively large errors. Therefore, developing efficient pre-processing approaches for spectral data to obtain higher prediction accuracy is in great demand.

In the process of spectral acquisition, spectra often exhibit signal overlap and noise interference, thus resulting in a decrease in resolution. Therefore, a signal processing algorithm for extracting target information effectively could improve the prediction accuracy. As a mathematical operation process, deconvolution is a typical signal extraction and recovery method [19,20], which showed unique advantages in spectral processing. The deconvolution technology improved the resolution beyond the instrument’s limit and significantly improved the signal-to-noise ratio [21,22]. To obtain an accurate and reliable signal, spectral deconvolution was associated with Gaussian fitting of the absorption spectrum to adjust the Gaussian mathematical curve and obtain corresponding characteristic absorption peaks from an overlapping spectrum.

In addition, the objective of this study was to pre-process the FTIR-ATR spectra of ^14^NO_3_^−^/^15^NO_3_^−^ mixture to obtain the characteristic absorption peaks by using a deconvolution algorithm, and a *PLSR* model based on the deconvoluted spectra was explored to predict nitrate nitrogen isotope, which could provide technical support for nitrate sources and water quality management.

## 2. Results and Discussion

### 2.1. Spectral Characterization

The mixed solutions of ^14^NO_3_^−^ and ^15^NO_3_^−^ with different proportions showed similar spectral characteristics (Figure 1). A total of two strong absorption peaks at 3800–3000 cm^−1^ and 1800–1500 cm^−1^ that were attributed to the characteristic absorption of water were observed. Although the characteristic peak of nitrate was at approximately 1500–1200 cm^−1^, the signal of the absorption peak was weak and was strongly interfered with by water, thus resulting in difficulty in direct observation.

### 2.2. Spectral Processing

The second derivative spectra of the of ^14^NO_3_^−^ and ^15^NO_3_^−^ mixtures with different ratios did not show prominent characteristic peaks in the range of 1500–1200 cm^−1^ (Figure 2a). The characteristic spectra of nitrate after directly deducting water exhibited an irregular trend (Figure 2b). One reason might be that the infrared-characteristic absorption of NO_3_^−^ was also strongly interfered with by water. Another factor could be attributed to the low concentration of NO_3_^−^ in the sample, resulting in a low characteristic peak intensity. Therefore, it is necessary to further process the spectra of the 1500–1200 cm^−1^ band to obtain target spectral information from the overlapping band.

The peak-fit 4.12 software was used to deconvolve the spectra of the 1500–1200 cm^−1^ band: the goodness of fitting (*R*^2^) obtained was all above 0.97, and the sum of squared errors (*SSE*) was all less than 0.108, exhibiting excellent deconvolution effects. The characteristic peak position and the overall intensity of ^14^NO_3_^−^ and ^15^NO_3_^−^ mixtures with different proportions were obviously different (Figure 2c). With the increase in the proportion of ^15^NO_3_^−^ in mixtures, the characteristic peak of nitrate gradually shifted in the direction of a low wavenumber. The nitrate absorption peaks were 1365.7, 1362.1, 1358.5, 1354.8, 1344.1, 1340.4, and 1336.8 cm^−1^ when the ^14^N/^15^N ratios were 1:0, 3:1, 2:1, 1:1, 1:2, 1:3, and 0:1, respectively. Additionally, the overall spectral intensity decreased with the increase in the proportion of ^15^NO_3_^−^, the main reason probably was that when the absorption peak of ^14^NO_3_^−^ and ^15^NO_3_^−^ mixture was close to the characteristic absorption of water (1800–1500 cm^−1^) stronger interference could be observed. As shown in Figure 2d, the proportion of ^15^NO_3_^−^ showed a good negative correlation with the wavenumber of the nitrate absorption peaks. Therefore, the proportion of ^15^NO_3_^−^was calculated based on the characteristic peak positions of the ^14^NO_3_^−^ and ^15^NO_3_^−^ mixtures. However, for different ^14^N/^15^N ratios, the characteristic peak intensity of nitrate was proportional to the nitrate concentration; therefore, the absorption peaks could be used for the quantitative analysis of nitrate in solutions.

### 2.3. Principal Component Analysis

The principal component analysis (PCA) based on the spectra range of 1500–1200 cm^−1^ was performed (Figure 3). In the water deduction algorithm and deconvolution algorithm, the first two principal components both accounted for more than 96% of the whole spectral variance. Therefore, PC1 and PC2 were used to represent the spectral variation. However, the score plot showed an irregular pattern with the water deduction algorithm (Figure 3a). Instead, it was found that the scores of these two principal components showed a regular and consistent distribution with the deconvolution algorithm (Figure 3b). The plot shifted to negative values of PC1 as the ratio of ^15^NO_3_^−^ increased, and the PC2 scores of all seven ^14^N/^15^N combinations shifted from negative values to positive values with the increase of total N concentrations, which further proved that FTIR-ATR coupled with deconvolution could be used to determine ^15^NO_3_^−^ in the mixed solutions of ^14^NO_3_^−^ and ^15^NO_3_^−^.

### 2.4. Prediction of Nitrate Nitrogen with a Water Deduction Algorithm

The *PLSR* was used to model the spectra of the 1500–1200 cm^−1^ region based on the water deduction algorithm (Figure 4). Firstly, the cross-validation method was performed to obtain the optimal number of principal components in the seven mixed solutions of ^14^NO_3_^−^ and ^15^NO_3_^−^. The optimal number of principal components were 6, 1, 3, 1, 1, 1 and 1 when the ^14^N/^15^N ratios were 1:0, 3:1, 2:1, 1:1, 1:2, 1:3, and 0:1, respectively, corresponding to the minimum value of the *RMSECV* (Figure 4a,c,e,g,i,k,m). Therefore, the optimal principal components were used to construct the *PLSR* model. The *R*^2^ between the measured and predicted values in the validation set were 0.722, 0.862, 0.772, 0.628, 0.923, 0.747, and 0.394, respectively, and the *RPD* values were 1.86, 2.15, 1.83, 1.53, 2.60, 1.93, and 1.28, respectively, which were higher than the minimum standard of 1.4 required for quantitative determination except for the *RPD* of ^15^NO_3_^−^ solution (^14^N/^15^N = 0:1). However, the *LODs* of seven models were 8.75, 7.36, 9.21, 16.56, 5.32, 9.17, and 17.71 mg/L, respectively, which exhibited poor performance for the determination of low nitrate nitrogen concentration.

A bias value close to zero indicates a low systematic error between the measured and predicted values. The biases of the models were shown in Table 1. According to the absolute value, the low systematic errors of seven models were 0.302, 0.232, 0.192, 0.269, 0.163, 0.156, and 0.298, respectively.

### 2.5. Prediction of Nitrate Nitrogen with a Deconvolution Algorithm

The deconvoluted spectra in the range of 1500–1200 cm^−1^ were involved in the *PLSR modeling* (Figure 5). The optimal principal components were 2, 3, 2, 4, 3, 3, and 2 when the ^14^N/^15^N ratios were 1:0, 3:1, 2:1, 1:1, 1:2, 1:3, and 0:1, respectively (Figure 5a,c,e,g,i,k,m). Therefore, the *PLSR* model could be established with each optimal principal component. The results showed that the *R*^2^ values between the measured and predicted values in the validation set were 0.935, 0.960, 0.976, 0.843, 0.908, 0.982, and 0.857, respectively, showing a good linear relationship (Figure 5b,d,f,h,j,l,n). The *RPD* values were respectively obtained at 2.93, 2.59, 6.27, 2.38, 3.19, 4.57, and 2.55, which were all higher than the minimum standard of 1.4 that was required for quantitative determination. The prediction capacity of all the above models reached a good level. In addition, the *PLSR* models in the calibration set also exhibited robust prediction performance (Table 2), and the *LOD* of seven models were 3.73, 4.67, 2.31, 4.86, 3.57, 2.79, and 5.62 mg/L, respectively, significantly lower than that of the *LOD* with the water deduction algorithm. Over the last several decades, a large number of sensing methods have been developed to detect nitrate nitrogen in water. Several have a high sensitivity with a good *LOD* but expensive instrumentation; others have a reasonable sensitivity with reduced cost. It has been reported that an ion chromatography technique can be useful to determine nitrate nitrogen in water with a good *LOD* of 0.05 mg/L [23], while the *LOD* of some detection methods based on biosensors and voltammetry was higher than 5 mg/L [24,25]. Although our analytical methods did not show excellent *LOD* compared to some reference methods, the *LOD* were acceptable for determination. In fact, in spectral analysis, *RPD* could also be used to represent the sensitivity. In all seven ^14^N/^15^N combinations, the *RPD* of the *PLSR* prediction models based on deconvolution was more than 2, indicating that the models were at a good level. However, compared to the models with the water deduction algorithm, all seven models for nitrate prediction showed lower systematic errors (Table 2). 

In previous studies, the *PLSR* models, coupled with the deconvolution algorithm, showed good prediction performance for low concentrations of ^14^NO_3_^−^ (Table 3). For the sample with a high ^15^NO_3_^−^ concentration, the models based on the water deduction algorithm were acceptable for prediction, but for the sample with a low ^15^NO_3_^−^ concentration, the spectra with water deduction showed strong interference and irregular trends, and the models exhibited poor prediction capacity (Figure 4). In this experiment, a deconvolution algorithm was adopted to pre-process the spectra to obtain an effective small target signal and further achieve rapid determination for low concentrations of ^14^NO_3_^−^ and ^15^NO_3_^−^. Overall, the above results indicated that this method could be used for the quantitative determination of nitrate in ^14^NO_3_^−^ and ^15^NO_3_^−^ mixtures with different proportions.

In this study, as a theoretical investigation, standard samples were prepared and used in the investigation. However, the model developed in a different aquatic environment might be different due to varied background interference on the spectra. Our previous study confirmed that the presence of carbonate in the water impacted the nitrate determination [15]. Therefore, when this technology is applied to different aquatic environments (e.g., surface water and groundwater) in the future, the factors that may affect the prediction performance of the models need to be fully considered and investigated. Further, the parameters of the spectral deconvolution algorithm can also be optimized to further improve the sensitivity and applicability of the model.

## 3. Materials and Methods

### 3.1. Materials

The test reagents were ^14^NO_3_-K (an analytical reagent, purchased from Nanjing Ronghua Apparatus Co., Ltd., Nanjing, China), and ^15^NO_3_-K with a ^15^N abundance of 99% (an analytical reagent, purchased from Shanghai Jizhi Biochemical Technology Co., Ltd., Shanghai, China). A total of two stock solutions of 250 mg/L ^14^NO_3_^−^ and 250 mg/L ^15^NO_3_^−^ were prepared, and then the standard mixture solutions with a total N concentration of 100 mg/L ^14^NO_3_^−^ and ^15^NO_3_^−^ at different ^14^N/^15^N ratios (1:0, 3:1, 2:1, 1:1, 1:2, 1:3, 0:1) were obtained by adjusting the amount of the two stock solutions of ^14^NO_3_^−^ and ^15^NO_3_^−^. Finally, the standard sample solutions had total N concentrations of 0, 1, 4, 8, 12, 16, 20, 24, and 28 mg/L at different ^14^N/^15^N, respectively. For the seven combinations of different ^14^N/^15^N ratios, three replicates were prepared at each concentration level.

### 3.2. Spectra Recording and Pre-Processing

A 4300 handheld FTIR spectrometer with an ATR spectra accessory (Thermo Fisher Scientific, Waltham, MA, USA) was used to record the spectra in the range of 4000 to 600 cm^−1^, with a spectral resolution of 4 cm^−1^, 32 scans. The atmospheric and instrumental noises were corrected by subtracting the background from each scan. An appropriate amount of each sample was placed in the sample tank for three measurements.

The FTIR-ATR spectra were prepossessed by the Savitzky–Golay smoothing filter to improve the signal-to-noise ratio by eliminating baseline floating and noise [26]. According to the absorption characteristics of nitrate, the spectrum in the 1500–1200 cm^−1^ region was subjected to smoothing, baseline correction, and Gaussian deconvolution with Peak-fit 4.12 software. The deconvolution process has been elaborated in detail in Figure 6.

The deconvolution of the spectral curve assumed that several single-peak spectral bands superimpose the experimental spectrum *Y*(*x*). The purpose of fitting was to find the single-peak spectral band *F_i_*(*x*) (*i* = 1, 2 …, *n*) [27,28]. The principle was as follows:*Y*(*x*) = *Σ**F_i_*(*x*)(1)
where *Y* is the spectrum, *x* is the wave number, *i* (1, 2, 3 …, *n*) is the number of independent peaks, and *F* is the kernel function of the expansion or deconvolution. The Gaussian function is taken as the kernel function:(2)$$y=\frac{a_{0}}{\pi \sqrt{\pi a_{2}}}exp\left[-\frac{1}{2}{\left(\frac{x-a_{1}}{a_{2}}\right)}^{2}\right]$$
where *a*_0_, *a*_1,_ and *a*_2_ represent peak amplitude, position, and width, respectively, while *x* and *y* represent wave number and absorption intensity.

### 3.3. Spectra Recording and Pre-Processing

The principal component analysis (PCA) is a dimensionality reduction chemometrics technique because it reduces redundant information in a data set. In this experiment, PCA was used to reduce the dimensions of the spectral data by providing new variables and to find the regularities among the different ratios of ^14^N/^15^N in the spectral subsection. The PCA treated the intense peak positions in each spectrum as vectors and formed linear combinations of the vectors by assigning a weight to each vector. The MATLAB software (MATLAB2018b, the MathWorks, Natick, MA, USA) was used for the data analysis.

### 3.4. Prediction Models

The partial least squares regression (*PLSR*) model is one of the most commonly used stoichiometry algorithms in spectral data analysis. It was a bilinear model where a matrix *X*, containing the variables (spectra wavenumber), and a matrix *Y*, a function of the variables in matrix *X* (nitrate contents), were used to predict the smallest number of latent variables. Cross-validation is a statistically sound method for choosing the number of components in the *PLSR* model. It avoids over-fitting data by not reusing the same data to both fit a model and estimate the prediction error. Thus, the estimate of the prediction error is not optimistically biased downward. *PLSR* has an option to estimate the root-mean-squared-error (*RMSE*) by cross-validation, and when the model reached the first lowest *RMSE*, the corresponding number of factors was optimal [29].

### 3.5. Calibration and Validation Datasets

The spectral data sets of each ratio of ^14^NO_3_^−^/^15^NO_3_^−^ mixture were randomly divided into a calibration data set containing 75% spectra and a validation data set containing the remaining 25% spectra.

### 3.6. Model Evaluation

The ratio of prediction to deviation (*RPD*), the correlation coefficient (*R*^2^), and the root-mean-square error (*RMSE*) were used to evaluate the model’s predictive performance. The parameters were evaluated by the following equations:(3)$$RMSE=\sqrt{\frac{1}{n}\overset{n}{\underset{i=1}{\sum }}{\left(y_{i}-{\hat{y}}_{i}\right)}^{2}}$$
(4)$$RPD=\frac{SD}{RMSE}$$
(5)$$R^{2}=1-\frac{\sum_{i=1}^{n}{\left(y_{i}-{\hat{y}}_{i}\right)}^{2}}{\sum_{i=1}^{n}{\left(y_{i}-\bar{y}\right)}^{2}}$$
where $y_{i}$ and ${\hat{y}}_{i}$ denote the measurement value and predicted value of nitrate in the *e*_th_ sample, respectively; $\bar{y}$ is the mean value of the measured nitrate content; *SD* is the standard deviation. *RMSE_C_* and *RMSE_V_* represent the root mean square error (*RMSE*) in the calibration and validation dataset models. High *R*^2^, *RPD*, and low *RMSE* define the robustness and accuracy of the models. *RPD* value less than 1.4 is considered unsuitable for quantitative measurement; *RPD* value between 1.4 and 1.8 is acceptable; *RPD* value between 1.8 and 2.0 is good, in which case quantitative prediction could be made; *RPD* value between 2.0 and 2.5 is very good quantitative analysis; and an *RPD* higher than 2.5 indicates excellent model prediction performance [30].

### 3.7. Limit of Detection 

In the process of linear multivariate analysis, the limit of detection (*LOD*) reflected the model’s sensitivity. The *LOD* was calculated by 3σ/m, where σ was the standard deviation of the predicted concentration, which was denoted by *RMSEP,* and *m* was the fitting-curve slope of the model (using real value as the X-axis and predicted value as the Y-axis) [31].

### 3.8. Systematic Error Assessment

The systematic errors are derived from the inadequacy of calibration models. The bias is the sum of the differences between the estimated value $y_{i}$ and the measured value $x_{i}$ [32,33], and was calculated by Equation (6):(6)$$bias=\frac{\sum \left(y_{i}-x_{i}\right)}{n}\;$$
where *n* is the number of samples. A bias value close to zero indicates a low systematic error between the measured and predicted values [34].

## 4. Conclusions

Combining the deconvolution algorithm and *PLSR* model, FTIR-ATR, was used to determine the N isotope in the ^14^NO_3_^−^ and ^15^NO_3_^−^ mixtures with different proportions. The wavenumber of nitrate absorption peaks exhibited a regular shift as the proportion of ^15^NO_3_^−^. The *PLSR* models showed good prediction performance of nitrate content in the mixed solutions of ^14^NO_3_^−^ and ^15^NO_3_^−^, which achieved the quantitative determination of ^15^NO_3_^−^ content. The above results implied that FTIR-ATR technique coupled with deconvolution algorithm was an effective and alternative option for estimating the isotopic composition of nitrate in water bodies, which provided a simple, rapid and affordable method for isotopic tracing of nitrate nitrogen, further generating a better understanding of nitrate sources and transformation processes.

## Figures and Tables

**Figure 1 molecules-28-00567-f001:**
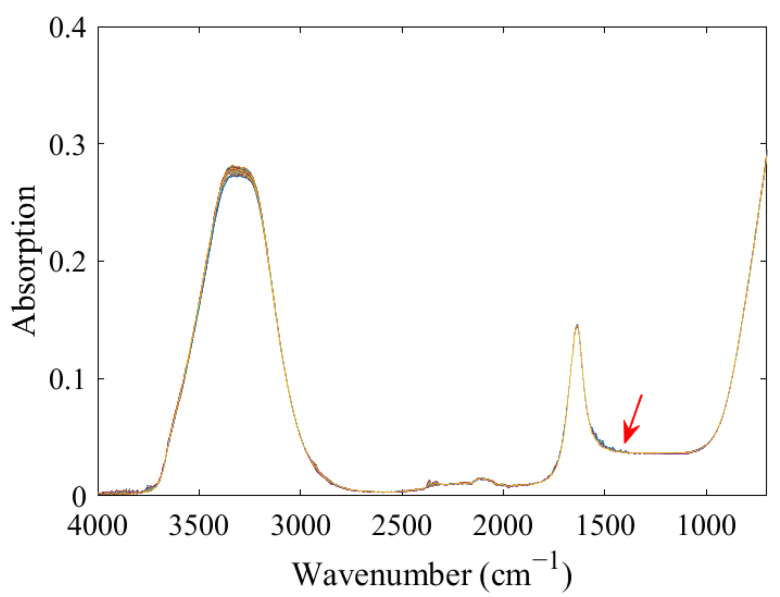
FTIR-ATR spectra of ^14^NO_3_^−^ and ^15^NO_3_^−^ mixtures with different proportions.

**Figure 2 molecules-28-00567-f002:**
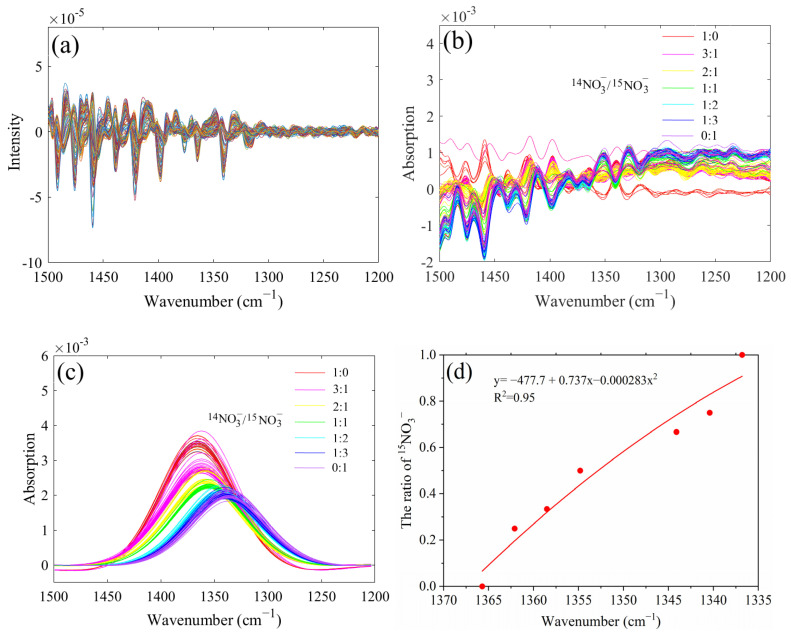
The second derivative spectra of nitrate in the range of 1500 to 1200 cm^−1^ (**a**), characteristic absorption bands of nitrate through water deduction (**b**), deconvoluted spectra of ^14^NO_3_^−^ and ^15^NO_3_^−^ mixtures with different proportions (**c**), the linear correlation between the proportion of ^15^NO_3_^−^ and the wavenumber of the nitrate absorption peaks (**d**).

**Figure 3 molecules-28-00567-f003:**
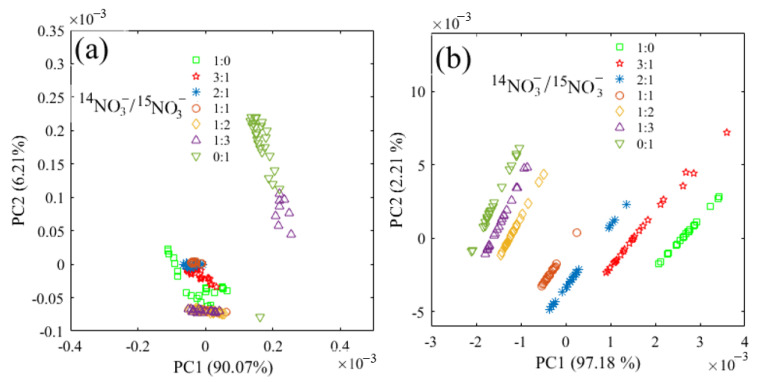
Principal component distribution based on the spectra in the range of 1500–1200 cm^−1^ for the nitrate mixture with different ^14^N/^15^N ratios. (**a**) water deduction algorithm; (**b**) deconvolution algorithm.

**Figure 4 molecules-28-00567-f004:**
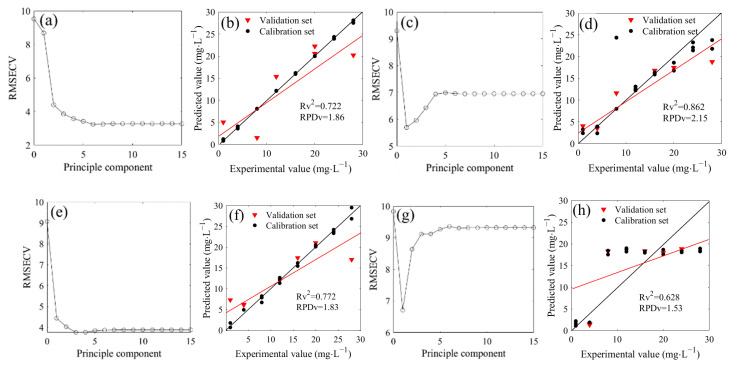
Distribution and model evaluation of the partial least squares regression (*PLSR*) principle component (**a**,**c**,**e**,**g**,i,**k**,**m**), training set (n = 18), and testing set ((**b**,**d**,**b**,**h**,**j**,**l**,**n**); n = 6) of the *PLSR* prediction model based on FTIR-ATR spectra (with water deduction algorithm) of of ^14^NO_3_^−^ and ^15^NO_3_^−^ mixtures with different proportions (^4^N/^15^N, 1:0 (**a**,**b**); 3:1 (**c**,**d**); 2:1 (**e**,**f**); 1:1 (**g**,**h**); 1:2 (**i**,**j**); 1:3 (**k**,**l**); 0:1 (**m**,**n**)).

**Figure 5 molecules-28-00567-f005:**
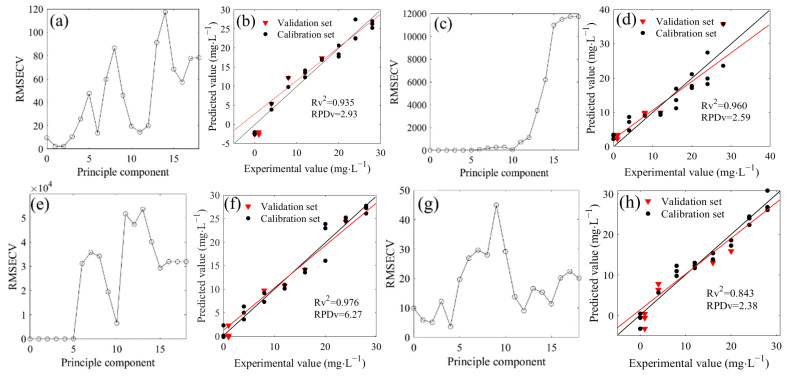
Distribution and model evaluation of the partial least squares regression (*PLSR*) principle component (**a**,**c**,**e**,**g**,**i**,**k**,**m**), training set (n = 20), and testing set ((**b**,**d**,**f**,**h**,**j**,**l**,**n**); n = 7) of the *PLSR* prediction model based on deconvoluted spectra of ^14^NO_3_^−^ and ^15^NO_3_^−^ mixtures with different proportions (^4^N/^15^N, 1:0 (**a**,**b**); 3:1 (**c**,**d**); 2:1 (**e**,**f**); 1:1 (**g**,**h**); 1:2 (**i**,**j**); 1:3 (**k**,**l**); 0:1 (**m**,**n**)).

**Figure 6 molecules-28-00567-f006:**
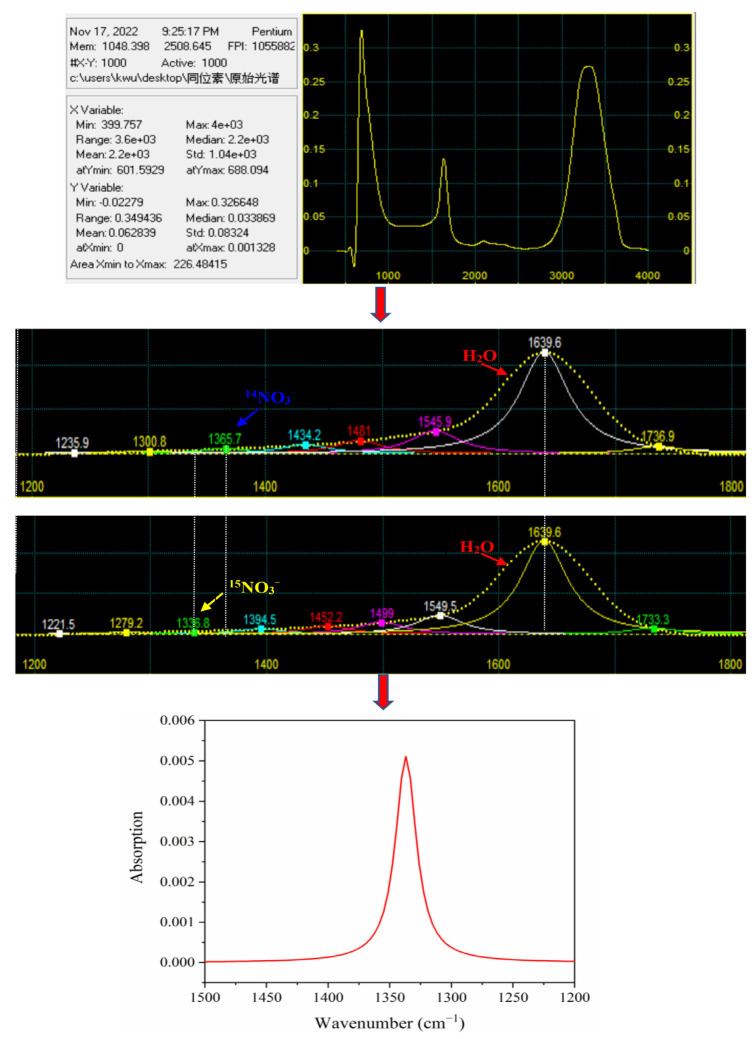
The process of spectra deconvolution.

**Table 1 molecules-28-00567-t001:** Statistics of the *PLSR* models used in the calibration and validation sets for the prediction of f ^14^NO_3_^−^ and ^15^NO_3_^−^ in the mixtures with different proportions using FTIR-ATR spectra (with water deduction algorithm).

^14^NO_3_^−^/^15^NO_3_^−^	Calibration			Validation			Bias
	*R_C_* ^2^	*RMSE* _ *C* _	*RPD* _ *C* _	*R_V_* ^2^	*RMSE_V_*	*RPD_V_*	
1:0	0.999	0.071	44.32	0.722	2.57	1.86	0.302
3:1	0.741	2.62	1.96	0.862	2.13	2.15	0.232
2:1	0.993	0.78	12.13	0.772	2.71	1.83	0.192
1:1	0.555	3.37	1.49	0.628	3.28	1.53	−0.269
1:2	0.900	1.75	3.17	0.923	1.61	2.60	0.163
1:3	0.867	1.98	2.74	0.747	2.55	1.93	−0.156
0:1	0.460	3.19	1.36	0.394	3.35	1.28	0.298

Notes: *PLSR*: partial least squares regression; *RMSE*: the root mean square error; *RPD*: the ratio of prediction to deviation.

**Table 2 molecules-28-00567-t002:** Statistics of the *PLSR* models used in the calibration and validation sets for the prediction of ^14^NO_3_^−^ and ^15^NO_3_^−^ in mixtures with different proportions using deconvoluted spectra.

^14^NO_3_^−^/^15^NO_3_^−^	Calibration			Validation			Bias
	*R_C_* ^2^	*RMSE_C_*	*RPD_C_*	*R_V_* ^2^	*RMSE_V_*	*RPD_V_*	
1:0	0.952	1.15	4.68	0.935	1.21	2.93	0.105
3:1	0.851	1.63	2.59	0.960	1.43	2.59	0.079
2:1	0.961	0.82	5.06	0.976	0.76	6.27	0.088
1:1	0.957	0.78	4.82	0.843	1.55	2.38	−0.062
1:2	0.971	0.71	5.85	0.908	1.12	3.19	0.041
1:3	0.952	0.82	4.56	0.982	0.85	4.57	−0.036
0:1	0.877	1.57	2.86	0.857	1.62	2.55	−0.031

Notes: *PLSR*: partial least squares regression; *RMSE*: the root mean square error; *RPD*: the ratio of prediction to deviation.

**Table 3 molecules-28-00567-t003:** Nitrate nitrogen isotope determination based on the *PLSR* model using FTIR-ATR spectroscopy.

Nitrate Nitrogen Isotope	Concentration	Spectral Pre-Processing	Statistic Parameters			References
			*R* ^2^	*RPD*	*RMSE*	
^14^NO_3_^−^	0–20 (mg·L^−1^)	Deconvolution	0.986	3.15	0.203	[14]
^14^NO_3_^−^	0–4.3 (mg·L^−1^)	Deconvolution	0.886	2.76	0.286	[15]
^15^NO_3_^−^	0–200 (mg·L^−1^)	Water deduction	0.990	4.85	9.20	[16]
^15^NO_3_^−^	0–120 (mg·kg^−1^)	Water deduction	0.980	8.15	3.91	[17]
^15^NO_3_^−^	0–200 (mg·L^−1^)	Water deduction	0.998	4.76	/	[18]

Notes: *RPD*: the ratio of prediction to deviation; *RMSE*: the root mean square error.

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
