# Peer review of "Rapid Determination of Nitrate Nitrogen Isotope in Water Using Fourier Transform Infrared Attenuated Total Reflectance Spectroscopy (FTIR-ATR) Coupled with Deconvolution Algorithm"

_molecules, 2023, doi:10.3390/molecules28020567_

Round 1
Reviewer 1 Report
This is a good effort to explore the potential of Fourier transform infrared attenuated total reflectance spectroscopy (FTIR-ATR) in combination with chemometrics for rapid detection of nitrate-nitrogen isotope in water. The approach is useful,therefore, this MS could be considered for publication in the journal of Molecules. However, authors need to address following points for improvement.
1. Has been applied this method over real samples? or all the samples analyzed were "made' in the lab?
2. In subsection 2.2, they talk about a type of filter (Savitzky – Golay), can the filter remove the signs of water? explain better, please.
3. The authors used Peakfit 4.12 software to conduct deconvolution, did the parameters of software remain the same for each spectrum?
4. Line 191, “Figure 2c” should be “Figure 3c”.
5. Line 268, “Figure 5” should be “Figure 6”.
Reviewer 2 Report
This study reports an FTIR-ATR spectroscopic method for the determination of nitrate-nitrogen isotope in Water. This study has been managed approximately well. In my opinion, this study can be published in the ‘’Molecule’’ after applying some revisions.
1 Since there are some similar studies in the literature, what is the novelty of this study? It should be explained.
2 In Lines 42-43, the statements of ammonia nitrogen (NH4 + - N), nitrate nitrogen (NO3 - - N) and nitrite nitrogen (NO2 - - N) are confused and the related part should be rewritten.
3 In lines 49, it seems that ‘’including’’ should be replaced with ‘’include’’.
4 The preparation of the mixture solutions of 14NO3 - and 15NO3 - with different 14N/15N ratios 102 (1:0, 3:1, 2:1, 1:1, 1:2, 1:3, 0:1) should be explained in detail.
5 The language of the manuscript should be checked thoroughly to avoid confusion.
6 The obtained results should be compared to the literature. It seems like a report in the present form. A table can be prepared for the comparisons. (Yan-Qiu SHAO, Chang-Wen DU, Ya-Zhen SHEN, Fei Ma, Jian-Min ZHOU, Chinese Journal of Analytical Chemistry, Volume 42, Issue 5, May 2014, Pages 747-752; Molecules 2020, 25(24), 5838; https://doi.org/10.3390/molecules25245838; Ke Wu, Fei Ma, Zhenwang Li, Cuilan Wei, Fangqun Gan, Changwen Du, Journal of Environmental Management, Volume 317, 1 September 2022, 115452; Yanqiu Shao, Changwen Du, Jianmin Zhou, Fei Ma, Ying Zhu, Kai Yang and Chao Tian, Anal. Methods, 2017,9, 5388-5394)
Reviewer 3 Report
This is an interesting article on the determination of nitrate-nitrogen isotope in water. The manuscript was well structured and concise with significant contribution. However, there are several concerns needing clarification and further elaboration from the authors as follows:
1. Title. Please standardize whether “nitrate-nitrogen” or “nitrate nitrogen” to be used in the manuscript.
2. 2.1 Materials. It was unclear how the total N concentrations were considered and prepared. Are these concentration levels prepared for all the seven combinations? How many replicates were prepared in this study?
3. 3.1 Spectral characterization. Line 169: It is better using the term “similar” rather than “same”.
4. 3.2 Spectral processing. Line 191: Figure 3c.
5. Line 195-197: “Additionally, the overall spectral intensity decreased with the increase of the proportion of 15NO31-, the main reason…… closer to… characteristic absorption of water, the stronger the interference.” From the understanding, the higher the proportion of 15N, the lower the wavenumber and further from the water region. Please verify.
6. Figure 3(d) – The relationship between the wavenumber and ratio seems to be a polynomial rather than a linear curve which need clarification from the authors.
7. 3.3 Principal component analysis. PCA was not described in methodology and suggested to be included.
8. 3.3 PCA. The shift to negative values for PC1 was evident in the figure. How about the trand of PC2?
9. 3.4 Prediction of nitrate nitrogen. Line 237-238. RPD of 15N (14N:15N = 0:1). Error in 14N.
10. Line 231 and line 267. How was the selection of PC performed for the prediction?
11. Line 239-244, 276-278. The procedure for determination of LOD shall be included in methodology. The author shall discuss on the LOD whether the method was adequate sensitive for the determination.
12. It was suggested to include the limitations of the study in the manuscript, as this method was not tested in solution with matrix effect. It was unclear how the matrix effect might interfere the results, especially when the method is intended to apply for water quality management as proposed.
